# Regression activation mapping on the cortical surface using graph convolutional networks

**Ben A. Duffy**[1]                                                    BDUFFY@INI.USC.EDU
**Mengting Liu**[1]                                                        MLIU@INI.USC.EDU
**Trevor Flynn**[2]                                              TREVOR.FLYNN@UCSF.EDU
**Arthur W. Toga**[1]                                                  TOGA@LONI.USC.EDU
**A. James Barkovich**[2]                                  JAMES.BARKOVICH@UCSF.EDU
**Duan Xu**[2]                                                        DUAN.XU@UCSF.EDU
**Hosung Kim**[1]                                                        HKIM@INI.USC.EDU
[1] *USC Stevens Neuroimaging and Informatics Institute*
[2] *University of California, San Francisco*

## Abstract

Saliency mapping on graph convolutional networks (GCNs) is important for model inter-pretability and has not yet been investigated for regression GCNs employing graph down-sampling. We examined regression activation mapping for localizing the salient regions identified by GCNs. We first demonstrated using simulations that it is possible to generate precise vertex-wise saliency maps on the cortical surface mesh and then applied this method to age prediction using cortical surfaces derived from neonatal structural MRI.

## 1. Introduction

Graphs are ubiquitous in neuroimaging e.g. for modelling connectivity or cortical surface meshes (Cucurull et al., 2018). Graph convolutional networks (GCNs) filter input features through a given graph topology. Identifying salient nodes is important for model inter-pretability and while this has been attempted before on classification tasks (Arslan et al., 2018), to our knowledge it has not been investigated for regression GCNs or GCNs employ-ing graph downsampling, a necessary building block for GCNs with small localized filters. Here, we investigate regression activation mapping (RAM) for localizing the salient regions identified by regression GCNs in the input graph space. We first demonstrate using simula-tions that it is possible to generate precise vertex-wise saliency maps on the cortical surface mesh and then apply this method to age prediction using cortical surfaces derived from neonatal structural MRI.

## 2. Methods

### 2.1. Graph CNN and Regression Activation Mapping

We used a spectral GCN approach (Defferrard et al., 2016), with graph convolutions de-fined as multiplications in the Fourier domain. The signal on the graph nodes is filtered by a $K$-localized filter defined by the first $K$ terms of the Chebyshev polynomial of the

graph Laplacian, where here we tested both $K=3$ and 4. Graph coarsening (pooling) was employed using the Graculus clustering algorithm (Dhillon et al., 2007; Defferrard et al., 2016). Feature maps in the final convolutional layer were weighted by the output layer weights and summed to generate RAMs (Zhou et al., 2016). After 3 pooling layers, RAMs are about 8 times coarser than the input graph and were therefore mapped back to the nodes on the original graph by assigning the node values of the children in the coarser graph to the nodes of the parent graph. RAMs explain which nodes in the final layer contribute to the output value, therefore the contribution at each node needs to be assessed relative to some reference. Here, we generate population-wise saliency maps using the node-wise standard deviation of the test data.

## 2.2. Neonatal Dataset and Cortical Surface Extraction

Our dataset comprised of 83 preterm infants without brain injury on MRI. Most subjects were scanned twice, providing 105 scans after quality control. The white/gray matter interface and pial surface were extracted using the method in (Kim et al., 2016). Surfaces were registered to a template and 2 morphological features, the cortical thickness and sulcal depth, were measured in native space on 81,924 vertices. To reduce computation time for GCN training, surfaces were downsampled to 1284 vertices using icosahedron sampling.

## 2.3. Simulated Dataset

To evaluate the method and GCN architecture against a known ground truth, a simulated dataset with 90 training samples and 10 test examples was generated by sampling from a normal distribution at each vertex. In order that output regression labels were of the same scale as the background, labels were drawn from this same distribution and placed onto 90 connected nodes on the input mesh (Fig. 1a). Following this, gaussian noise was added with variances equal to 0.2, 0.4, 0.8, 1.6 and 3.2 times the original signal variance. Saliency mapping methods should be sensitive to training the model on a random permutation of the labels (Adebayo et al., 2018), therefore we employed this test on the simulated dataset.

## 2.4. Training and Testing of GCNs

We used a GC8-P2-GC16-P2-GC32-P2-GC64-GAP architecture, where GC$n$ is a graph convolutional layer with $n$ features, P is a pooling layer and GAP is a global average pooling layer. ReLU activations were used after each GC layer. A Mean squared error loss function was optimized using stochastic gradient descent with a momentum of 0.9, a batch size of 8 and a learning rate of $5\times10^{-5}$ that was decayed by 0.95 every 1000 steps. The network was trained for 20000 iterations, which took about 2 min on a NVIDIA 1080Ti GPU. For age prediction, a 5-fold cross validation (CV) grouped by subject was run 5 times for each parameter set. Performance was evaluated using cortical thickness and sulcal depth together as well as each feature separately.

## 3. Results

The saliency maps generated using RAM accurately recovered the ground-truth signal nodes (Fig. 1a), despite 3 graph coarsening layers. Quantitatively, GCNs trained using data with

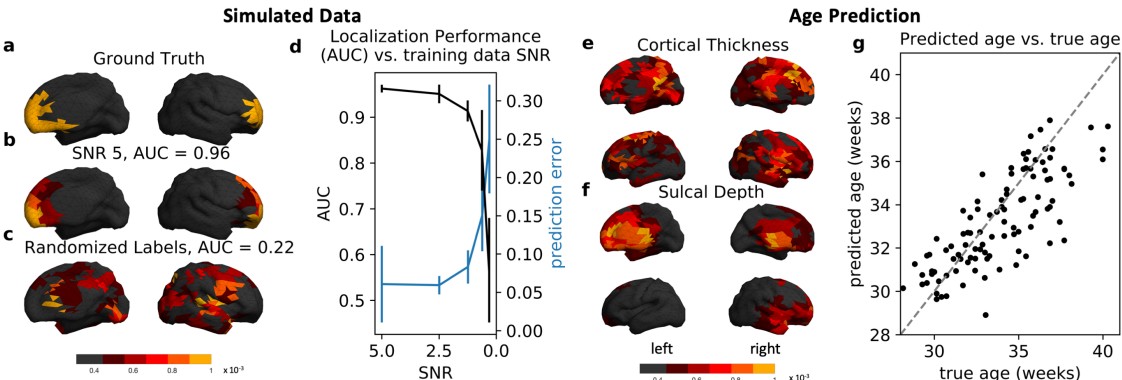

Figure 1: Saliency mapping on simulated data (a-d) and age prediction task (e-g). (a) The output labels were inserted onto 90 connected nodes on the cortical mesh. Example saliency maps for: (b) simulated test data after training the GCN on training data with an SNR of 5, (c) test data after training with permuted labels for SNR=5. (d) Localization performance vs. increasing noise level (error bars represent SD over 10 simulation runs). Saliency map for models: (e) trained using cortical thickness alone and (f) trained using sulcal depth alone. (g) Predicted age vs. true age for model trained using both features.

an SNR of 5 yielded a mean AUC of 0.96 (Fig. 1b-d). The method was robust to noise and achieved a mean AUC of 0.82 at an SNR of 0.6 (Fig. 1d). Finally, activation mapping was highly sensitive to training the model with randomized labels and in this case yielded a mean AUC of 0.5±0.2 (Fig. 1c).

Contrary to (Arslan et al., 2018), for our age prediction task we found pooling improved predictive performance for both $K=3$ (MAE=1.37 vs. 1.41 weeks, p<0.05, t-test) and $K=4$ (MAE=1.37 vs. 1.40 weeks, p<0.05). Higher values of $K$ were not found to improve performance and would likely hurt localization precision. Saliency maps for cortical thickness highlighted the right insular, right posterior cingulate and left anterior cingulate cortices (Fig. 1e), whereas sulcal depth was most important in the right anterior and posterior cingulate cortices and the left parahippocampal gyrus (Fig. 1f). The regional importance of cortical thickness was mapped over most cortical areas whereas that of sulcal depth was mapped preferentially on the anterior part of the brain. Training with cortical thickness and sulcal depth improved the predictive performance compared to training with thickness or sulcal depth alone (CV MAE = 1.37 vs. 1.80/2.15 weeks, Fig. 1g).

## 4. Conclusions

Here, we demonstrated the utility of regression activation mapping on GCNs using simulations and cortical surface data. Our simulations established that the method accurately recovers the ground truth signal despite noisy input while at the same time the method passed the data randomization test. Application of the method to age prediction highlighted important regions for age prediction in preterm infants, some of which are known as the areas of rapid growth in the 3rd trimester of gestation.

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
