# OpenReview forum: "Regression activation mapping on the cortical surface using graph convolutional networks"
_MIDL.io/2019/Conference/Abstract — MIDL Abstract 2019_

### Official Review · AnonReviewer2 · 2019-04-30
**Extension of related work to an age regression problem**

**Rating:** 3
**Confidence:** 2

**Review:**

This paper proposes a method for age prediction from cortical surfaces using graph convolutions. Furthermore, saliency maps are investigated. This work is a straight-forward extension of Arslan et al. to a regression problem. It does not contain substantial methodological novelties, but rather presents a proof of concept on a new task using an established method.

I do not completely follow the synthetic experiments. Why were regression labels "placed onto 90 connected nodes"? Shouldn't each mesh have a single regression label to make it analogous to the age regression problem?

---

### Official Review · AnonReviewer1 · 2019-05-02

**Rating:** 3
**Confidence:** 2

**Review:**

The paper talks about the RAM on GCNs using simulations and real data. Through simulation, the authors were able to show that the saliency mapping generated was consistent with the ground truth and was robust to noise.
The proposed method then used in age prediction problem and help explain the high-impact region.

I would suggest the authors to improve the organization of this abstract and clarify the workflow. For now, the abstract is mixing methods, data and experiment setup all together. This fact makes it hard to follow.

---

### Decision · Program_Chairs · 2019-05-06
**Acceptance Decision**

Accept